# IgG antibody responses to *Anopheles gambiae* gSG6-P1 salivary peptide are induced in human populations exposed to secondary malaria vectors in forest areas in Cameroon

Cyrille Ndo[1,2,3☯]*, Emmanuel Elanga-Ndille[2,4☯]*, Glwadys Cheteug[5,6], Rosine Danale Metitsi[7], Samuel Wanji[5], Carole Else Eboumbou Moukoko[1,6,8]*

1 Department of Biological Sciences, Faculty of Medicine and Pharmaceutical Sciences, University of Douala, Douala, Cameroun, 2 Department of Medical Entomology, Centre for Research in Infectious Diseases, Yaounde, Cameroon, 3 Organisation de Coordination pour la Lutte Contre les Endemies en Afrique Centrale, Malaria Research Laboratory, Yaounde, Cameroon, 4 Vector Borne Diseases Laboratory of the Biology and Applied Ecology Research Unit, Department of Animal Biology, Faculty of Sciences, University of Dschang, Dschang, Cameroon, 5 Department of Microbiology and Parasitology, University of Buea, Buea, Cameroon, 6 Malaria Research Unit, Centre Pasteur Cameroon, Yaounde, Cameroon, 7 Department of Animal Biology and Physiology, University of Yaounde 1, Yaounde, Cameroon, 8 Laboratory of Parasitology, Mycology and Virology, Postgraduate Training Unit for Health Sciences, Postgraduate School for Pure and Applied Sciences, University of Douala, Douala, Cameroon

☯ These authors contributed equally to this work.
* emmsdille@yahhoo.fr (EEN); elsecarole@yahoo.fr, eboumbou@pasteur-yaounde.org (CEEM); cyrndo@yahoo.fr (CN)

**Data Availability Statement:** All data underlying the findings described in the paper are fully

## Abstract

Human IgG antibody response to *Anopheles gambiae* gSG6-P1 salivary peptide was reported to be a pertinent indicator for assessing human exposure to mosquito bites and evaluating the risk of malaria transmission as well as the effectiveness of vector control strategies. However, the applicability of this marker to measure malaria transmission risk where human populations are mostly bitten by secondary vectors in Africa has not yet been evaluated. In this study, we aimed to investigate whether anti-gSG6-P1 antibodies response could be induced in humans living in forest areas in Cameroon where *An. gambiae s.l* is not predominant. In October 2019 at the pick of the rainy season, blood samples were collected from people living in the Nyabessang in the forest area in the South region of Cameroon. Malaria infection was determined using thick blood smear microscopy and Rapid Diagnostic Test. The level of IgG Anti-gSG6-P1 response as a biomarker of human exposure *to Anopheles* bite, was assessed using enzyme-linked immunosorbent assay. Mosquitoes were collected using the human landing catches to assess *Anopheles* density and for the identification of Anopheles species present in that area. IgG antibody response to the gSG6-P1 salivary peptide was detected in inhabitants of Nyabessang with high inter-individual heterogeneity. No significant variation in the level of this immune response was observed according to age and gender. The concentration of gSG6-P1 antibodies was significantly correlated with the malaria infection status and, *Plasmodium falciparum*-infected individuals presented a significantly higher level of IgG response than uninfected individuals (p = 0.0087). No significant difference was observed according to the use of insecticide

available without restriction and presented in the Supporting information files. The data file is also uploaded in the public repository Dryad, Dataset, https://doi.org/10.5061/dryad.kprr4xh7q.

**Funding:** The authors received no specific funding for this work.

**Competing interests:** The authors have declared that no competing interests exist.

**Abbreviations:** An., Anopheles; ELISA, Enzyme-linked immunosorbent assays; gsg6-P1, gambiae salivary gland protein 6-peptide 1; HLC, Human landing catches; IgG, Immunoglobulin G; ITNs, Insecticide-treated nets; mRDTs, Malaria rapid diagnostic tests; NMCP, National Malaria Control Program; WHO, World health organization; ΔOD, Delta optical density.

treated nets. Out of the 1,442 *Anopheles* mosquitoes species collected, 849 (58.9%) were identified as *An. paludis*, 489 (33.91%) as *An. moucheti*, 28 (4.44%) as *An. nili*, 22 (2.08%) as *An. gambiae s.l* and 10 (0.69%) as *An. marshallii*. Our findings show that IgG response to *An. gambiae* gSG6-P1 peptide could be detected in humans exposed predominantly to *An. moucheti* and *An. paludis* bites. Taken together, the data revealed the potential of the Anti-gSG6-P1 IgG antibody response to serve as a universal marker to assess human exposure to any *Anopheles* species.

## Introduction

During the two last decades, the massive use of vector control strategies such as Long Lasting Insecticide Nets (LLINs) and Indoor Residual Sprays (IRS) has led to a drastic decrease in malaria morbidity and mortality, worldwide [1]. These results have motivated the WHO to envision the reduction of malaria transmission by at least 90% by 2030 [2, 3]. To achieve this goal, strategies are mainly based on intensification of the use of vector control strategies, especially in the most affected African countries such as Cameroon [4, 5]. In the case that these strategies continue to sustainably reduce human–vector contact as expected, endemic areas will be stratified with foci of residual transmission and the challenge will undoubtedly be to quantify changes in transmission intensity over space and time. Traditionally, the entomological measure of malaria transmission intensity is mainly based on the estimation of the entomological inoculation rate (EIR) [6]. However, this indicator which depends on both the biting and sporozoite infection rates can be challenging to be estimated in low transmission settings, where the density of mosquitoes can be extremely low [7, 8]. Given the relevance of vector surveillance data in assessing malaria transmission intensity, any tools that could provide more accurate data would be of great value.

Recently, human IgG antibodies specific response to one salivary peptide from the *An. gambiae* salivary Gland Protein-6 (the gSG6-P1 salivary peptide) was abundantly reported to be significantly associated with recent exposure to *An. gambiae* s.l and *An. funestus* s.l mosquitoes species in several transmission settings in Africa [9–14]. Furthermore, the level of immune response to the gSG6-P1 salivary peptide was reported to be negatively associated with the efficient use of LLINs in Africa [15, 16] indicating that this marker could detect a drop in human exposure to *Anopheles* mosquito bites due to the implementation of control measures. Moreover, specific immune response to the gSG6-P1 peptide was also observed to be correlated with infection to *Plasmodium falciparum* parasite in humans [17] ascertaining recent exposure to the bites of *Anopheles* mosquitoes.

The gSG6-P1 salivary peptide of *An. gambiae* was also found to be applicable in other non-African malaria-endemic settings including South America [18], Southeast Asia [19] and Oceania [20]. However, the fact that the IgG Ab response to this peptide was not found to be associated with human exposure to *An. faurati* bites in the Solomon Islands [21] further question whether and/or to which extent the anti-gSG6-P1 antibody levels might serve as an accurate proxy to estimate human recent exposure to anopheline bites. It would therefore be recommendable that the usefulness of IgG Ab response to gSG6-P1 peptide as a marker of exposition to malaria vectors bites be investigated in epidemiological settings where other *Anopheles* species than *An. gambiae s.l* and *An. funestus s.l* are the dominant species. This is particularly the case in forested areas of South Cameroon where species such as *An. moucheti*, *An. nili* and *An. paludis* are significantly more abundant and play an important role in malaria transmission

[22–24]. The present study investigated the potential of anti-gSG6-P1 antibodies response as a marker of human exposure to the bites of these malaria vectors.

## Materials and methods

### Study design and population

This study was carried out during a pre-intervention survey as part of a pilot project aiming to assess the impact of house improvement in reducing the exposure of human populations to the bites of malaria vectors in the locality of Nyabessang (2˚24' North and 10˚24' East), one of the sentinel sites for malaria entomological surveillance in Cameroon. This village of the AMBAM health district in the South region is located in the Congo-Guinean phytogeographic area and is characterized by a typical equatorial climate with two rainy seasons from March to June and from September to November with the average annual rainfall varies between 1600 and 1800 mm [26]. This area is populated by around 240 inhabitants and the main activities carried out by the population are essentially hunting, fishing and agriculture. The village is surrounded by water bodies including three rivers (Ntem, Biwoume and Ndjo'o) and an artificial lake created during the construction of the Memve'ele hydroelectric dam. This later activity also considerably modified the landscape of the village with a considerable reduction in vegetation cover.

The populations of the study site were sensitized a week earlier and invited to participate in the study. Individuals included in this study were recruited among inhabitants of each of the 10 houses used for adults mosquito collection during the entomological survey. This is justified by the fact that, with the purpose to generate less biased information on the effectiveness of house improvement for reducing malaria transmission, the project aimed to compare only the data collected (entomological and epidemiological) from the inhabitants of only the 10 houses included in the study before and after the intervention. However, the minimum number of inhabitants expected for each of the 10 houses had to be greater than or equal to three. So, the minimum size of our sample in total was 30, which is a number often estimated as being sufficient to conduct significant statistics [25]. Furthermore, in each house, participants who met the eligibility conditions and agreed to participate in the study were recruited. Eligibility for inclusion was defined as a resident who had not travelled out of the study site within the last 3 weeks. Pregnant women, children under 6 months and individuals with severe clinical signs of malaria as defined by WHO were excluded.

The entomological survey (*Anopheles* mosquito density and species composition) was assessed in October 2019 at the pick of the rainy season and two weeks before the parasitological survey. Data collection sheets were used to collect data on socio-demographic characteristics (house geographic GPS location, age, gender,), and on the use of malaria control strategies (drug use, possession and use of LLINs). Parasitological and serological data were also obtained.

### Ethical approval and consent to participate

This study was conducted following ethics directives related to research on humans in Cameroon. The protocol of the study received from the approval of the Regional Ethical Committee for Research in Human Health of South Cameroon (N ˚ 006 / CRERSH SUD / SE/2019). Before their enrolment in the study, populations were first informed on the objectives and processes of the study (background, goals, methodology, study constraints, data confidentiality, and rights to opt out from the study. Thereafter, signed informed consent was obtained from all those who agreed to participate in the study following the Helsinki Declaration. Participation was voluntary, anonymous and without compensation.

## Adult mosquitoes collections and field processing

Adult female mosquitoes were collected using the human landing catches (HLC) technique from 06:00 PM to 06:00 AM indoors and outdoors of human dwellings. Mosquito collections were done by 20 adults volunteers (up to 18 years old) recruited in the village during two consecutive nights in the 10 houses randomly selected throughout the village. Al volunteers freely received antimalarial prevention before experiments, as per the national guidelines for malaria management. For each house, one collector was installed indoors and another one outdoors. The collectors exposed part of their legs and arms and, when they felt landing mosquitoes, they turned on a torch and collected the mosquitoes by inverting a small glass tube over it. Each tube was transferred into labelled bags according to collection time. Collected *Anopheles* species were subsequently sorted by species according to the morphological identification keys of Gillies and De Meillon [26] and Gillies and Coetzee [27]. Each mosquito was then preserved in 1.5 ml Eppendorf tubes kept in a plastic container containing silica gel, transported to the insectary of OCEAC and kept in– 20˚C until further analyses.

## Blood sampling and parasitological analysis

Two weeks after entomology survey, three drops of blood were collected from the fingertips of each individual by a qualified medical technician, and used for: (i) malaria rapid diagnostic test (SD Bioline Malaria Ag *P.f*/Pan Ag), (ii) thick blood smears for microscopic diagnosis and (iii) dried blood spot (DBS) on filter paper (3M) for the quantification of the level of human anti-gSG6-P1 IgG antibody response.

*P. falciparum* parasitaemia was determined by microscopic examination of Giemsa-stained thick blood smears. Parasite density was determined based on the number of parasites per 200 leukocytes on a thick film, assuming total leukocyte counts of 8,000 cells/microL of whole blood. All positive patients were free treated with the combination artemether-lumefantrine in accordance with the treatment guidelines from the Cameroon National Malaria Control Program.

## Anti-*Anopheles* gSG6-P1 IgG antibody detection by enzyme-linked immunosorbent assay

The recombinant *Anopheles*-specific salivary peptide gSG6-P1 peptide previously described by Poinsignon in 2008 [9], was synthesized and purified (> 95%) by Genepep SA (Montpellier, France). All peptide batches shipped in lyophilized form were subsequently suspended in ultrapure water and stored in aliquots at -20˚C until use.

The level of IgG Ab response to gSG6-P1 salivary peptide was measured on whole blood eluates obtained from standardized dried blood spots (Whatman 3M) according to a protocol previously described by Drame *et al.*, [28]. Briefly, 96-well Maxisorp plates (Nunc, Roskilde, Denmark) were pre-coated with 100 microL/well of gSG6-P1 salivary peptide solution (20 μg/ml of antigen in 100 microL of PBS 1m, pH 7.4) and incubated for 2h30min at 37˚C. Plates were then blocked with 300 microL/well of Protein-Free PBS Blocking-Buffer, pH 7.4 (Pierce, Thermo Scientific, France) for 45 minutes at 37˚C. Each eluate blood (1/50 dilution in PBS-Tween 1%) was incubated at 4˚C overnight in triplicate in the microplates (2 wells with antigen and called "Ag$^+$" and one well without antigen and called "Ag$^-$"). A volume of 100 microL /well of Biotin Mouse Anti-Human IgG the secondary Ab solution (BD Pharmingen$^{TM}$) diluted at 1/2000 in PBS-Tween 1% buffer was added following 1h30min of incubation at 37˚C. Streptavidin-Peroxidase conjugate (GE Healthcare UK) was then added at 1/2000 dilution and incubated for 1h at 37˚C with 100 μL/well. Finally, 100 microL/well of ABTS

(2,2-azino-bis (3-ethylbenzthiazoline 6-sulfonic acid) diammonium; Thermo scientific) substrate solution (0,05 M citrate buffer, pH 4) containing 10 microL of oxygenate water-$H_2O_2$ (30%) were added and incubated for 2 hours in the dark at room temperature for the colorimetric development and the optical density (OD) were read at 405 nm.

The level of IgG Ab response to gSG6-P1 antigen for each individual was expressed as the ΔOD value: ΔOD = ODx—ODn, where ODx represents the mean of individual OD in both antigen wells and ODn the individual OD in the well without gSG6-P1 antigen (to remove in each sample all non-specific reaction and background). For each sample, the experiment was validated only if the coefficient of variation between the two $Ag^+$ well (%CV) was <20%. Samples with CV >20% were re-analysed.

IgG Ab responses to the salivary peptide was also measured on 34 individuals from France (European volunteer blood donors with no travel history to malaria-endemic countries and with no history of exposure to Anopheles mosquitoes) and who served as negative controls.

All participants with ΔOD > 0.20 (cut-off) were defined as immune responders to the gSG6-P1 salivary peptide. The cut-off value was defined as a mean ΔOD of negative control plus three times the standard deviation (SD).

## Statistical analysis

Fisher's exact test was used to compare qualitative variables. After verifying that specific IgG response data (expressed in ΔOD) did not assume Gaussian distribution, the nonparametric Mann–Whitney U test was used to compare antibody levels between two independent groups, and the nonparametric Kruskal–Wallis test was used to compare more than two independent groups. Only p <0.05 values were considered significant. All statistical analyses were performed using GraphPad Prism5 software (San Diego, CA, USA) and Stata software (version 11 SE).

## Results

### Mosquito species composition in Nyabessang

Overall, 2,039 mosquitoes were collected and female Anophelines accounted for 70.72% (1,442/2,039) of mosquitoes. Among them, five species were identified; *An. paludis and An. moucheti* were the most important and represent 92.8% (1338) of the *Anopheles* samples (Table 1). *An. gambiae* was found in 2% of samples. *Anopheles* mosquitoes were mostly collected outdoors (54.2%) than indoors (45.8%).

### Malaria infection status

A total of 51 participants aged from 1 to 80 years old (Mean = 21.56± 21.40; 95%CI: 15.6–27.6) and living in the 10 households selected for mosquito collection, were recruited for the study. Among them, 32 were less than 21 years old (62.74%) whereas, 19 (37.25%) were more than 21

**Table 1.** *Anophele*s mosquito species composition in Nyabessang during the study period in October 2019.

| Species collected | Indoor (*n* = 660) | Outdoor (*n* = 782) | Total (*n* = 1,442) | Proportion (%) |
|---|---|---|---|---|
| *An. gambiae sl* | 22 | 8 | 30 | 2.08 |
| *An. marshallii* | 3 | 7 | 10 | 0.69 |
| *An. moucheti* | 231 | 258 | 489 | 33.91 |
| *An. nili* | 28 | 36 | 64 | 4.44 |
| *An. paludis* | 376 | 473 | 849 | 58.87 |

years old. With a proportion of 62.8%, females were the predominant gender within the studied population.

The majority (70.6%; 36/51) of participants declared to frequently use insecticide bed nets each night to protect against mosquitoes bites. Among them, 55.5% (20/36) were children aged from 0 to 15 years. No active fever case was recorded among participants. However, 39.2% (20/51) of participants were infected with *Plasmodium* parasite as detected by RDT, 21.6% by microscopy and 21.57% by both RDT and Microscopy. *Plasmodium falciparum* was identified in all microscopy-positive individuals and 3 (27.27%) of them had a co-infection with *Plasmodium malariae*. Children aged less than 14 years old account for 60% of the *Plasmodium*-infected population with 30% of them aged from 0 to 5 years old.

## IgG Ab response to *Anopheles* gSG6-P1 peptide in human population from Nyabessang

IgG Ab response specific to the *An. gambiae* gSG6-P1 salivary peptide was detected in human individuals living in malaria endemic area of Nyabessang (median ΔOD: 1.995, 95%CI: 1.948–2.368). The level of this specific response was significantly higher in Nyabessan population compared to European (median = 0.2256, IC95% = 0.1944–0.3000); P < 0.0001 non-parametric Mann-Whitney U-test) with the cut-off value (ΔOD = 0.20) used as negative controls (Fig 1). This immune response was highly heterogeneous between individuals and no significant difference was observed according to sex and age.

## Level of IgG Ab responses to *Anopheles* gSG6-P1 peptide according to malaria infection status

Comparison of the levels of IgG Ab response between *Plasmodium*-infected and uninfected individuals showed that this response was significantly higher (P = 0.0087, non-parametric Mann–Whitney test) in the group of infected individuals (median = 2.486; 95%CI: 2.141–2.841) than in uninfected ones (median = 1.940; 95%CI: 1.691–2.189; Fig 2).

To test the hypothesis that IgG-gSG6-P1 antibody level might serve as an indicator to estimate the probability to be infected with *P. falciparum*, participants were divided into two sub-groups around the value of the median level of anti-gSG6-P1 IgG Ab response (ΔOD = 1.995): (i) higher immune responders (with a ΔOD value ≥1.995) and (ii) lower immune responders (with a ΔOD value <1.995). The comparison between the two sub-groups showed that the probability to be found infected with malaria parasite was significantly higher (OR = 4.03, 95% CI: 2.200–7.385, P < 0.0001, Fisher's Exact Test) for "higher immune responders" than "lower immune responders" (Fig 3).

## Levels of IgG Ab responses to *Anopheles* gSG6-P1 antigen according to insecticide bed nets use

To investigate if the levels of specific IgG Ab responses to gSG6-P1 salivary peptide could constitute a tool for evaluating the impact of preventive measures against mosquito bites in areas where vectors other than *An. gambiae* s.l and/or *An. funestus* s.l are predominant, we compared the level of IgG Ab response between individuals who declared to frequently use the ITNs each night and those who do not. All participants who declared that they use ITNs had the bed nets installed in their bedrooms. No significant difference was observed in the levels of anti-gSG6-P1 peptide response in individuals that reported using ITNs and those that did not (Median = 2.187, 95%CI: 2.023–2.547 and 1.862 95%CI: 1.529–2.196, p = 0.0864, non-parametric Mann-Whitney U-test) (Fig 4).

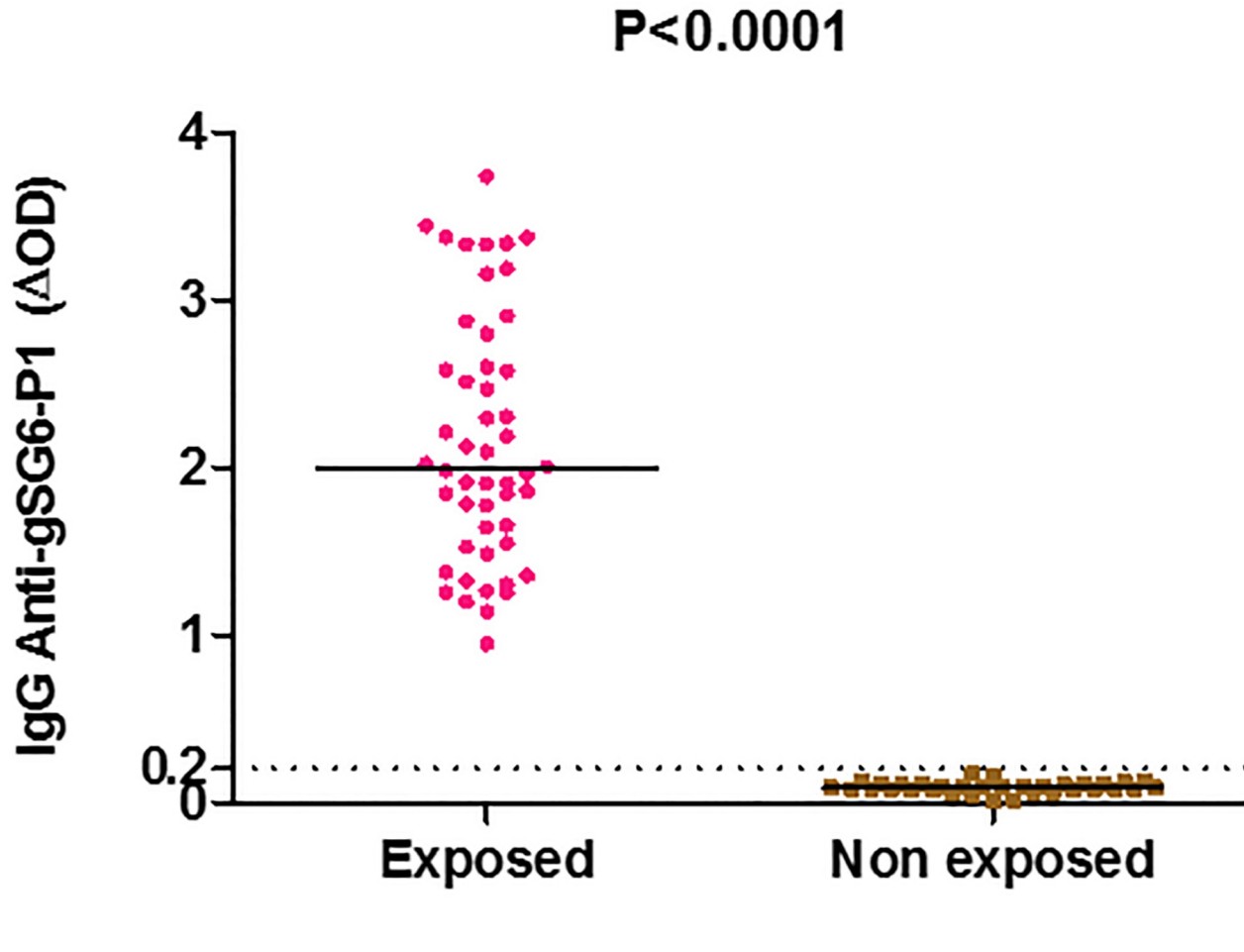

**Fig 1. IgG Ab response to the gSG6-P1 salivary peptide in individuals living in Nyabessang in a forest area in Cameroon.** Dot plots indicate the individual specific IgG level (ΔOD value) and bars represent the median value in each group. The black dotted line represents the cut-off of IgG response. The p-value was calculated using the Mann-Whitney U test.

## Discussion

Over the past decades, numerous studies have reported that specific immune responses to the *An. gambiae* s. l. gSG6-P1 salivary peptide could constitute a relevant tool to assess human exposure to *Anopheles* mosquito bites and consequently, the risk of malaria transmission and the efficacy of vector control strategies [9, 15–17, 29, 30]. In the present study, we observed that the gSG6-P1 salivary peptide was strongly recognized by blood samples collected from the human population predominantly exposed to the bite of vectors other than *An. gambiae* s.l and/or *An. funestus* s.l in forest areas in Central Africa. The predominance of exposure to these other vectors is demonstrated by data collected during the entomological surveys showing very low densities of *An. gambiae* s.l compared to those of *An. paludis* and *An. moucheti*. Similar vector distribution was also reported in another study conducted in this area by Bamou et al [22].

Specific immune response to the gSG6-P1 was previously reported to be an indicator of human exposure to not only *An. gambiae* s.l, but also to other vectors such as *An. funestus*

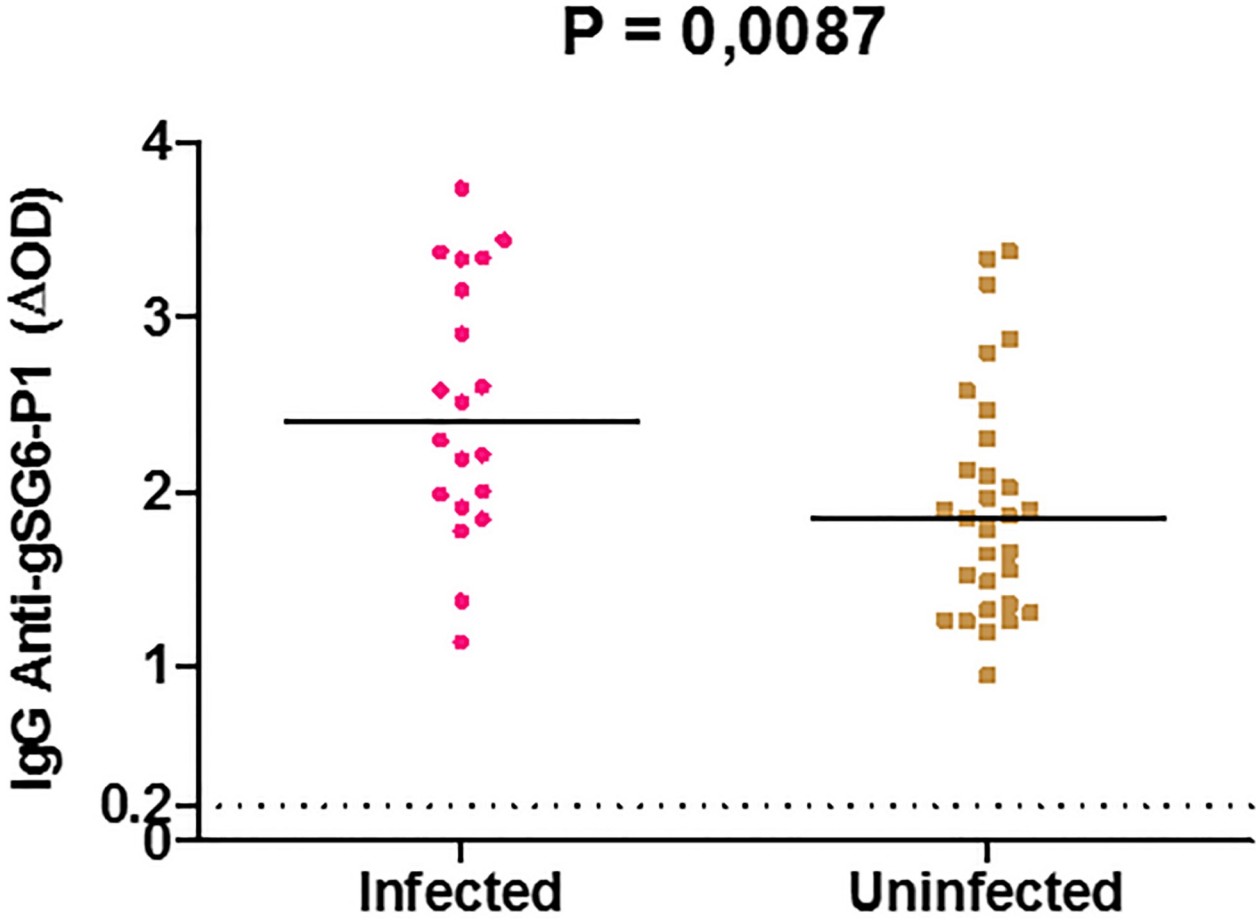

**Fig 2. IgG anti-gSG6-P1 peptide antibody levels according to malaria infection status in Nyabessang human population.** Specific IgG responses are shown (ΔOD) for *Plasmodium*-infected and uninfected individuals, bars indicate the median value for each group. The Cut-off is represented by the dotted line. *P*-value denotes the significance by the non-parametric Mann-Whitney U-test.

[30], American endemic *Anopheles* species [18] and Southeast Asia vectors [19]. The reactivity to the bite of such different vectors has been explained by the high conservation of the gSG6 protein between numerous *Anopheles* mosquito species [9, 31]. Thus, the detection of anti-gSG6-P1 IgG Ab responses in humans largely exposed to *An. moucheti* and *An. paludis* likely indicate a strong cross-reactivity between the gSG6 proteins members of these species and the one of *An. gambiae* s.l. This observation may not be surprising given that the SG6 proteins from malaria vectors of the subgenus Cellia (including *An. moucheti*, [32]) were reported to share 80% to 84% identity [31]. In the same way, it has been reported a highly conserved folding and a relatively high identity (60–65%) between *An. gambiae* s.l. SG6 protein and the ones of members of the *Anopheles* subgenus (which include *An. paludis* [32]) [33]. Thus, although no information is available on the sequences similarity between *An. gambiae* SG6 and its homologous from African secondary malaria vectors, it can be assumed that the recognition of gSG6-P1 peptide observed in this study is cross-reactivity with SG6 proteins from *An.*

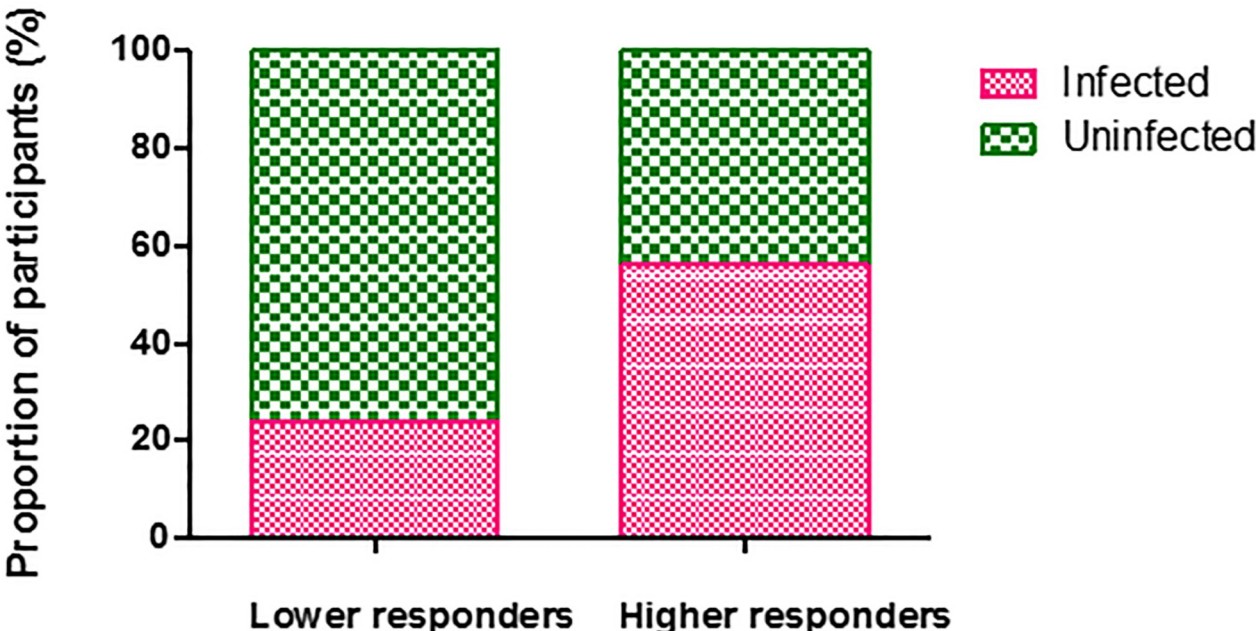

**Fig 3. Risk evaluation of malaria using IgG antibody levels against the gSG6-P1 peptide.** Boxes show the proportion of malaria infected individuals according to the level of responders to IgG l anti-gSG6-P1 characterized as lower and higher responders.

*moucheti* and *An. paludis*, the two most abundant *Anopheles* mosquito species in the study area. This highlights evidence that the human antibody response to *An. gambiae* gSG6-P1 salivary peptide might be detected in areas where human populations are largely bitten by vectors other than *An. gambiae* s.l and/or *An. funestus* s.l and reinforce the fact that this peptide is shared between different species of *Anopheles*, instead of focusing on *Anopheles gambiae*.

IgG Ab response to gSG6-P1 salivary peptide detected in this study in human individuals exposed to malaria secondary vectors showed high inter-individual heterogeneity indicating a different levels of exposure to vectors bites between individuals as observed in several studies [9, 28–30, 34]. This shows that IgG Ab response to gSG6-P1 could efficiently discriminate the level of exposure to the bites between individuals even if they are less bitten by *An. gambiae*. Furthermore, no difference in specific IgG response to the gSG6-P1 peptide was found according to gender. This result indicates that there is no influence of gender on the immune response specific to this salivary peptide. This is in line with previous studies reporting that the Ab response to gSG6-P1 peptide could detect human exposure to *Anopheles* bites regardless the gender [18, 19, 29, 34, 35] even if the level of IgG anti-gSG6-P1 seems to be little higher on female than male as we also observed. The same observation was made for the age emphasizing the suitability of the IgG Anti-gSG6-P1 response to be applied in all populations.

The level of IgG Ab response to gSG6-P1 peptide was significantly higher in infected participants than in uninfected ones. Our findings also showed that higher antibody levels are significantly associated with a higher possibility of having been bitten by a *Plasmodium*-infected mosquito. This suggests that the level of gSG6-P1 specific IgG could be a proxy for estimating the risk of acquiring malaria in areas where people are mostly bitten by secondary vectors as previously reported for exposure to the bite of major malaria vectors [14, 17, 18]. In the context of reducing malaria by at least 90% by 2030 as envisaged by WHO in 2015 such an indicator would be helpful for accurate assessment of the risk of transmission and the efficacy of control

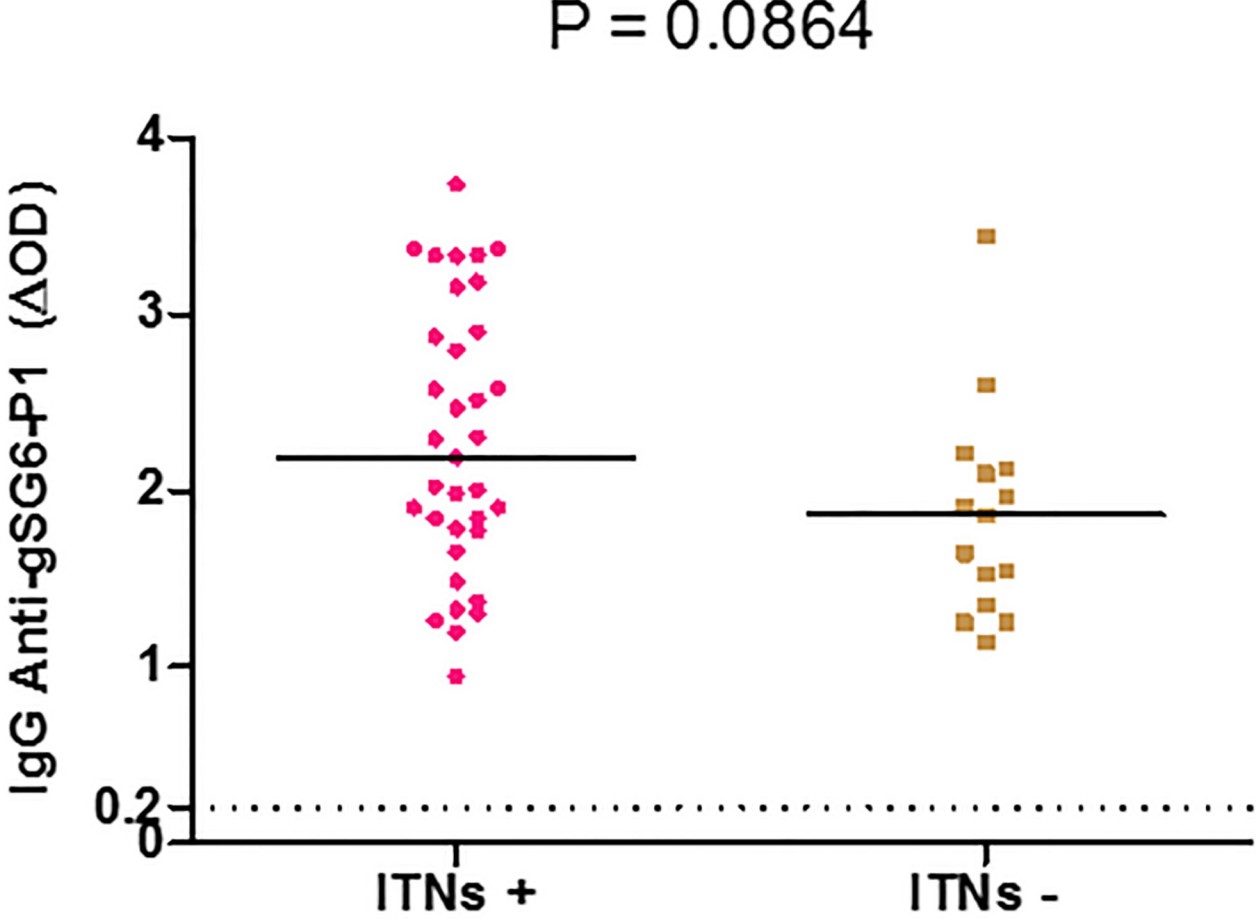

**Fig 4. IgG response to the gSG6-P1 peptide according to the use of an insecticide-treated net.** Dot plots show the level of individual specific IgG response (ΔOD value) to the gSG6-P1 peptide. Bars indicate the median value for each group. The cut-off is represented by the black dotted line.

interventions, especially in places where methods traditionally used such as, entomological inoculation rate and parasite prevalence are reported to be limited [7, 8].

Several previous studies have suggested that the IgG response against gSG6-P1 is a reliable alternative to accurately assess the effectiveness of malaria control methods. Indeed, these studies showed that the use of insecticide-treated bednets and spray bombs drastically reduced the level of IgG specific response to the gSG6-P1 peptide [15, 16]. In the present study, we found no significant influence of the use of insecticide-treated nets on the level of immune response to gSG6-P1 since no difference was observed between participants using bednets and those who do not. This absence of efficacy of ITNs might be related to vectors biting behavior in the study area. *Anopheles moucheti* and *An. paludis*, the two predominant vectors collected during the study can bite both indoor and outdoor in the study area [22, 24]. The potential effectiveness of ITNs would therefore be thwarted by the bites that the human would have received outdoor when they are out of bednet. However, we cannot rule out potential biases in

the collection of information on ITNs use since this was done based only on the declaration of participants.

## Conclusion

In the present study, we showed that the gSG6-P1 serologic biomarker is capable to detect human exposure to *An. moucheti* and *An. paludis*, two dominant species believe to be malaria vectors in the forest areas in Africa. Our findings highlight the pertinence of this biomarker to be used to assess malaria risk in all transmission settings in Cameroon. Such indicators could help the national malaria control programs accurately measure the effectiveness of their control interventions.

## Supporting information

**S1 Database.**
(XLSX)

## Acknowledgments

We are very grateful to the participants who agreed to participate in this study as well as the technical staff of the participating health facilities and the Malaria Research service for their support and cooperation during the survey.

## Author Contributions

**Conceptualization:** Cyrille Ndo, Emmanuel Elanga-Ndille, Samuel Wanji, Carole Else Eboumbou Moukoko.

**Data curation:** Cyrille Ndo, Emmanuel Elanga-Ndille, Glwadys Cheteug, Rosine Danale Metitsi, Carole Else Eboumbou Moukoko.

**Formal analysis:** Emmanuel Elanga-Ndille, Glwadys Cheteug.

**Funding acquisition:** Cyrille Ndo, Carole Else Eboumbou Moukoko.

**Investigation:** Cyrille Ndo, Emmanuel Elanga-Ndille, Rosine Danale Metitsi.

**Methodology:** Cyrille Ndo, Emmanuel Elanga-Ndille, Carole Else Eboumbou Moukoko.

**Project administration:** Cyrille Ndo, Carole Else Eboumbou Moukoko.

**Supervision:** Emmanuel Elanga-Ndille.

**Validation:** Cyrille Ndo, Emmanuel Elanga-Ndille, Glwadys Cheteug, Samuel Wanji, Carole Else Eboumbou Moukoko.

**Visualization:** Cyrille Ndo, Emmanuel Elanga-Ndille.

**Writing – original draft:** Emmanuel Elanga-Ndille, Glwadys Cheteug.

**Writing – review & editing:** Cyrille Ndo, Emmanuel Elanga-Ndille, Glwadys Cheteug, Rosine Danale Metitsi, Samuel Wanji, Carole Else Eboumbou Moukoko.

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
