## [Decision Letter · Decision Letter 0]

23 Aug 2022

PONE-D-22-13036IgG Antibody responses to Anopheles gambiae gSG6-P1 salivary peptide are induced in human populations exposed to secondary malaria vectors in forest areas in CameroonPLOS ONE

Dear Dr. N'DILLE,

Thank you for submitting your manuscript to PLOS ONE. After careful consideration, we feel that it has merit but does not fully meet PLOS ONE’s publication criteria as it currently stands. Therefore, we invite you to submit a revised version of the manuscript that addresses the points raised during the review process.

We look forward to receiving your revised manuscript.

Kind regards,

Anupkumar R. Anvikar, M.D.

Academic Editor

PLOS ONE

Journal Requirements:

2. In your Methods section, please provide additional information on how the sample size was determined and why sample size calculation was not carried out before sample collection.

Reviewers' comments:

Reviewer's Responses to Questions

**Comments to the Author**

1. Is the manuscript technically sound, and do the data support the conclusions?

Reviewer #1: Yes

Reviewer #2: Yes

2. Has the statistical analysis been performed appropriately and rigorously? 

Reviewer #1: No

Reviewer #2: Yes

3. Have the authors made all data underlying the findings in their manuscript fully available?

Reviewer #1: No

Reviewer #2: Yes

4. Is the manuscript presented in an intelligible fashion and written in standard English?

Reviewer #1: Yes

Reviewer #2: Yes

5. Review Comments to the Author

Reviewer #1: dear, please revised and incorporate all of the manuscripts again and revised them and check the writing, grammar, and plagiarism. if you do all this the articles become the best and give sound to readers.

Reviewer #2: The research paper entitled “IgG Antibody responses to Anopheles gambiae gSG6-P1 salivary peptide are induced in human populations exposed to secondary malaria vectors in forest areas in Cameroon” has described that antibody response to Anopheles gambiae gSG6-P1 salivary peptide exists among the inhabitants of Nyabessang, Cameroon, where Anopheles gambiae population is minimal, significant correlation between the concentration of gSG6-P1 antibodies with the malaria infection status and no significant difference to the use of insecticide treated nets-indicating outdoor transmission.

Comments

1. Line 38: Thanks to the …….. The sentence is to be modified

2. Line 189: The species name should start with lower case ( An. Paludis and An. Moucheti)

3. Table : The figures in Indoor and Total column is to be corrected

4. Line 220: the word significantly is duplicated and may be deleted

6. PLOS authors have the option to publish the peer review history of their article (what does this mean?). If published, this will include your full peer review and any attached files.

Reviewer #1: **Yes: **Anmut Assemie

Reviewer #2: **Yes: **Dr Manoranjan Ranjit

---

## [Author Response · Author response to Decision Letter 0]

6 Oct 2022

Reviewer #1: dear, please revised and incorporate all of the manuscripts again and revised them and check the writing, grammar, and plagiarism. if you do all this the articles become the best and give sound to readers.

Comment 1. Abstract means the short summary of the whole body but this does not show such summary. 

The Abstract should:

Describe the main objective(s) of the study

Explain how the study was done, including any model organisms used, without methodological detail.

Summarize the most important results and their significance. 

So please follow such concepts.

Answer:

We thank the reviewer for the remarks. This study is talking about the use of IgG anti- gSG6-P1 salivary peptide as immune biomarker to assess human exposure to others malaria vector bite than Anopheles gambiae. The main objective his well defined as you can read from line 36 to line 38: “”In this study, we aimed to investigate whether anti-gSG6-P1 antibodies response could be induced in humans living in forest areas in Cameroon where An. gambiae s.l is not predominant.”. Moreover, we clearly indicated the most important results of our study as well as their significance for malaria surveillance ( see line 36 - 38). 

However, it is true that we did not mention any methodology in the abstract. He has been corrected and the abstract was rewritten as follows: Human IgG antibody response to Anopheles gambiae gSG6-P1 salivary peptide was reported to be a pertinent indicator for assessing human exposure to mosquito bites and evaluating the risk of malaria transmission as well as the effectiveness of vector control strategies. However, the applicability of this marker to measure malaria transmission risk where human populations are mostly bitten by secondary vectors in Africa has not yet been evaluated. In this study, we aimed to investigate whether anti-gSG6-P1 antibodies response could be induced in humans living in forest areas in Cameroon where An. gambiae s.l is not predominant. In October 2019 at the pick of the rainy season, blood samples were collected from people living in the Nyabessang in the forest area in the South region of Cameroon. Malaria infection was determined using thick blood smear microscopy and Rapid Diagnostic Test. The level of IgG Anti-gSG6-P1 response was assessed using enzyme-linked immunosorbent assay. Mosquitoes were collected using the human landing catches to assess Anopheles density species composition in that area. IgG antibody response to the gSG6-P1 salivary peptide was detected in inhabitants of Nyabessang with high inter-individual heterogeneity. No significant variation in the level of this immune response was observed according to age and gender. The concentration of gSG6-P1 antibodies was significantly correlated with the malaria infection status and, Plasmodium falciparum-infected individuals presented a significantly higher level of IgG response than uninfected individuals (p=0.0087). No significant difference was observed according to the use of insecticide treated nets. Out of the 1,442 Anopheles mosquitoes species collected, 849 (58.9%) were identified as An. paludis, 489 (33.91%) as An. moucheti, 28 (4.44%) as An. nili, 22 (2.08%) as An. gambiae s.l and 10 ( 0.69%) as An. marshallii. Our findings show that IgG response to An. gambiae gSG6-P1 peptide could be detected in humans exposed predominantly to An. moucheti and An. paludis bites. Taken together, the data revealed the potential of the Anti-gSG6-P1 IgG antibody response to serve as a universal marker to assess human exposure to any Anopheles species. (See lines 32-55)

2. Line 38: Thanks to ELISA assays, remove it

Answer:

This was misplace and we remove it as suggested by the reviewer. The sentence was rewrite as follow: IgG antibody response to the gSG6-P1 salivary peptide was detected in inhabitants of Nyabessang with high inter-individual heterogeneity. (See lines 43-44)

3. General comments for Introduction: 

1. Please relate introduction to the titles

2. Show the problem of the Anopheles gambiae (s.l.) in your study area and the value of the study for the future. 

3. Indicate annual malaria case in the study area 

In general, the introduction should:

Provide background that puts the manuscript into context and allows readers outside the field to understand the purpose and significance of the study

Define the problem addressed and why it is important

Include a brief review of the key literature

Note any relevant controversies or disagreements in the field

Conclude with a brief statement of the overall aim of the work and a comment about whether that aim was achieved

Answer:

As mentioned earlier, our study discusses the use of anopheles salivary biomaker as an effective tool to assess human exposure to the bites of malaria vectors. The main objective of this work was to assess the level of IgG antibody response to the salivary peptide of Anopheles gambiae (IgG anti-gSG6-P1) in an area where Anopheles gambiae is not the abundant malaria vector. 

In our introduction, we have gone from the general to the specific by recalling first the WHO goal of eliminating malaria by 2030. Among the methods used to evaluate the effectiveness of the strategies put in place, we have the evaluation of the level of human-vector contact and this latter is usually done using entomological method. However, this entomological method have some limitation. It is used only at the community or population level, but do not give the actual level of exposure to the individual and information about the heterogeneity of individual exposure to Anopheles bites. Based on that, an immune epidemiological biomarker (IgG anti-gSG6-P1) has been developed and validated over the world as a tool to assess the level of human exposure to Anopheles bites at the population and individual level. This biomarker was validated and applied in area where An. gambiae is the predominant malaria vector and also in area where An. gambiae sl is less abundant or absent. However, in Central Africa and specifically in Cameroon, there was no information about the use of this biomarker to evaluate the level of human exposure to other malaria vector bites than An. gambiae. And this study aimed to investigate whether anti-gSG6-P1 antibodies response could be induced in humans living in forest areas in Cameroon where An. gambiae s.l is not predominant.

So, since all these information are provided in detail in our introduction, we did not clearly get the concern the reviewer about the structure of this introduction.

4. Mandatory data

1. Description of study area

 -geographical location 

 - annual rainfall, temperature, rainy seasons

 - study place 

2. how can Anopheles gambiae s.l identified ?

3. how you know female An. gambiae s.l? 

4. simply give blood for female An. gambiae s.l is possible way to get egg 

 5. From where you collect female An. gambiae s.l?

6. how many female An. gambiae s.l you collect for this study? And how many egg lay each female An. gambiae s.l ? 

7. how to measure the environmental effect on female An. gambiae s.l for your study? 

10. show the way of rearing clearly 

= generally this method miss so many information and it can not correlate with the result. Now I say the method that you use is not true/ scientific.

Answer:

All the details requested by the reviewer for the description of the study site were indicated in the submitted manuscript (see lines 102 - 113). 

Also, all details concerning the entomological survey were clearly indicated in the methodology section of the submitted manuscript (see lines 142 - 154). Globally it’s clearly said in the manuscript that entomological survey was carried out to identify the different malaria vectors circulating in the study site. For this purpose, adult female mosquitoes were collected by the human landing catch method. No larvae collection was made and none mosquito was reared in the lab. So, we do not clearly understand why the reviewer is talking about the eggs laying and mosquito rearing. HLC is a collection that allow to collect only female mosquitoes that come to bite human, so the present study wanted only to identify which anopheles species is present in Nyabessang and which one can bite human. No information on the environemental effect on Anopheles gambiae sl was collected because this was not the purpose of the study. 

Concerning the number of Anopheles gambiae s.l collected, this too is clearly indicated in the submitted manuscript (see line 219 in the table). Overall, only 30 individuals of this species were collected during the study. 

5. Line 155: Blood sampling and parasitological analysis, Who collect the blood ?

Answer:

This comment of the reviewer was taking into account and the person who collected the blood is now indicated in the revised manuscript as follows: Two weeks after entomology survey, three drops of blood were collected from the fingertips of each individual by a qualified medical technician, and used for: (i) malaria rapid diagnostic test (SD Bioline Malaria Ag P.f/Pan Ag), (ii) thick blood smears for microscopic diagnosis and (iii) dried blood spot (DBS) on filter paper (3M) for the quantification of the level of human anti-gSG6-P1 IgG antibody response. (See lines 156-166)

6. Line 212: Mosquito species composition in Nyabessang, How this result related with the title ? is it mandatory ? 

Answer:

This part of the study is very important and strongly related with the title of the study. The purpose of describing the Anopheles species composition in the study site was to confirm that Anopheles gambiae s..l is not the predominant vector in Nyabessang. This result strengthens the title of the study by showing that human population in Nyabessang are more exposed to the bite of An. paludis and An. moucheti than to An. gambiae s.l. This confirms as already observed in several different studies that human responses to the An. gambiae gSG6-P1 salivary peptide could be induced in human exposed to other malaria vectors than An. gambiae s.l.

7. Line 234: IgG Ab response to Anopheles gSG6-P1 peptide in human population from Nyabessan, Please include the figure and stastical analysis 

Answer:

We thank the reviewer for this suggestion, however, according to PLOSONE submission guideline figures are not put directly into the manuscript, but rather attached to the PLOS ONE’s website at the time of submission. It is only at the time of publication of the article that all the figures and supplementary data include figures will be put together in one document. However, you could have access to the figures, by clicking on the link from PLOSONE that you received, and you can download the manuscript and also the figures of the article.

8. Line 254-255: (An. Albimanus, An. darlingi, An. punctipenni, An. quadrimaculatus, and An. punctimacula), The reviewer put question mark ?????????

Answer:

As you can read from before on line 254 to line 255 : “”Specific immune response to the gSG6-P1 was previously reported to be an indicator of human exposure to not only An. gambiae s.l, but also to other vectors such as An. funestus [29], American endemic Anopheles species An. Albimanus, An. darlingi, An. punctipenni, An. quadrimaculatus, and An. punctimacula [18] and Southeast Asia vectors An. minimus, An. dirus, An. maculatus [19]””. 

All these species were listed to indicate the species found in these countries and for which the salivary biomarker works well. however, as the mention of these seemed to confuse the understanding of the document, we thought it best to remove them from the final manuscript. The sentence has been rewritten as follows: Specific immune response to the gSG6-P1 was previously reported to be an indicator of human exposure to not only An. gambiae s.l, but also to other vectors such as An. funestus [30], American endemic Anopheles species [18] and Southeast Asia vectors [19]. (See line 292 – 294)

9. Line 264: (which include An. paludis [31]) [32]. There is a reviewer question mark about the parenthesis between reference 31 and reference 32

Answer:

The brackets here is to specify the genus Anopheles paludis and the accompanying reference 31 (now 32). However, reference 32 (now 33) which comes after the brackets is to quote the entire sentence with respect to the fact that other anopheles have sequences homologous to that of Anopheles gambiae.. So, the sentence must be read as a whole to understand its meaning as follows : In the same way, it has been reported a highly conserved folding and a relatively high identity (60-65%) between An. gambiae s.l. SG6 protein and the ones of members of the Anopheles subgenus (which include An. paludis [32]) [33]. (See lines 300-302)

Reviewer #2: The research paper entitled “IgG Antibody responses to Anopheles gambiae gSG6-P1 salivary peptide are induced in human populations exposed to secondary malaria vectors in forest areas in Cameroon” has described that antibody response to Anopheles gambiae gSG6-P1 salivary peptide exists among the inhabitants of Nyabessang, Cameroon, where Anopheles gambiae population is minimal, significant correlation between the concentration of gSG6-P1 antibodies with the malaria infection status and no significant difference to the use of insecticide treated nets-indicating outdoor transmission.

Comments

1. Line 38: Thanks to the …….. The sentence is to be modified

Answer:

The recommendation has been taken into account and the sentence has been rewritten as followed: IgG antibody response to the gSG6-P1 salivary peptide was detected in inhabitants of Nyabessang with high inter-individual heterogeneity. (See lines 43 - 44)

2. Line 189: The species name should start with lower case (An. Paludis and An. Moucheti)

Answer:

Thank you for the remark, it was a mistake and we correct it as indicated: An. paludis and An. moucheti. (See lines 214) 

3. Table: The figures in Indoor and Total column is to be corrected

Answer: 

It has been done; We thank the reviewer for this observation. The mistake is coming from the number of An. marshallii in the Indoor column. Instead of 5 An. marshallii it is 3 specimens that was collected. (See in the table, line 219) 

4. Line 220: the word significantly is duplicated and may be deleted 

Answer:

The word significantly was duplicated as observed by the reviewer and has been deleted. The sentence was rewritten as follow: Comparison of the levels of IgG Ab response between Plasmodium-infected and uninfected individuals showed that this response was significantly higher (P = 0.0087, non-parametric Mann–Whitney test) in the group of infected individuals (median =2.486; 95%CI: 2.141-2.841) than in uninfected ones (median=1.940; 95%CI: 1.691-2.189; Fig 2). (See lines 249 - 251)

---

## [Editor Report · Decision Letter 1]

18 Oct 2022

IgG Antibody responses to Anopheles gambiae gSG6-P1 salivary peptide are induced in human populations exposed to secondary malaria vectors in forest areas in Cameroon

PONE-D-22-13036R1

Dear Dr. N'DILLE,

We’re pleased to inform you that your manuscript has been judged scientifically suitable for publication and will be formally accepted for publication once it meets all outstanding technical requirements.

Kind regards,

Anupkumar R. Anvikar, M.D.

Academic Editor

PLOS ONE
---

## [Editor Report · Acceptance letter]

3 Nov 2022

PONE-D-22-13036R1 

IgG Antibody responses to *Anopheles gambiae* gSG6-P1 salivary peptide are induced in human populations exposed to secondary malaria vectors in forest areas in Cameroon 

Dear Dr. Elanga-Ndille:

I'm pleased to inform you that your manuscript has been deemed suitable for publication in PLOS ONE. Congratulations! Your manuscript is now with our production department. 

Kind regards, 

on behalf of

Dr. Anupkumar R. Anvikar 

Academic Editor

PLOS ONE